Evidence of microplastics in water and commercial fish from a high-altitude mountain lake (Lake Titicaca)

Loayza Erick erickzender.loayzatorrico@ugent.be erickz.loayzatorrico@gmail.com 1 2
Trigoso Barrientos Amaya C. 2
Janssens Geert P.J. 1
1 Department of Veterinary and Biosciences, Faculty of Veterinary Medicine, Ghent University , Merelbeke , Belgium
2 Unidad de Ecología Acuática, Instituto de Ecología, Facultad de Ciencias Puras y Naturales, Universidad Mayor de San Andrés , La Paz , Bolivia
Oehlmann Jörg
Electronic publication date: 2022 Nov 9
Publication date: 2022
Volume: 10
Electronic Location ID: e14112
Received 2022 Feb 17; Accepted 2022 Sep 2
Copyright: ©2022 Loayza et al.
Copyright year: 2022
Copyright holder: Loayza et al.
License: This is an open access article distributed under the terms of the Creative Commons Attribution License, which permits unrestricted use, distribution, reproduction and adaptation in any medium and for any purpose provided that it is properly attributed. For attribution, the original author(s), title, publication source (PeerJ) and either DOI or URL of the article must be cited.
License URL: https://creativecommons.org/licenses/by/4.0/

Keywords: Fishery impacts, Fibers, Plastic pollution, Orestias

Funding: VLIR-UOS of Belgium BO2019TEA480A103 Complementary Study on the Fisheries Diagnosis of Lake Titicaca in Bolivia (UNDP) GIRH-TDPS Project, under the supervision of the Instituto del Mar del Perú (IMARPE) This work was supported by the project “Working for a healthier aquaculture in Lake Titicaca” funded by VLIR-UOS of Belgium (BO2019TEA480A103), and the Complementary Study on the Fisheries Diagnosis of Lake Titicaca in Bolivia (UNDP), within the GIRH-TDPS Project, under the supervision of the Instituto del Mar del Perú (IMARPE). The funders had no role in study design, data collection and analysis, decision to publish, or preparation of the manuscript.

==============================
Microplastic pollution is a widespread environmental concern. Like other anthropogenic pollutants, microplastics can reach aquatic ecosystems through rivers and interact with the aquatic biota. For instance, Lake Titicaca (between Bolivia and Peru), one of the great ancient lakes in South America (3,809 m a.s.l.), shows a pollution problem, particularly in the southern shallow basin (Lago Menor) in Bolivia. Nevertheless, our knowledge of the presence of microplastics and their interaction with the biota of Lake Titicaca is limited. Therefore, this study evaluated the presence of microplastics in the stomach content of the four fish species targeted by local fisheries in Lago Menor of Lake Titicaca (Orestias luteus, Orestias agassizii, Trichomycterus dispar, and Odonthestes bonariensis; N = 1,283), and looked for relationships with trophic guilds or fishing areas. Additionally, surface water was analyzed to evaluate the presence of microplastics in the water. The evaluation of microplastics was carried out by visual observations. We observed that the frequency of microplastic ingestion was low in all species (<5%). Conversely, microplastic was present in the water, with the highest quantity at the southern part of Lago Menor (103 ± 20 particles per L), without differences in the microplastic number between sites. Most microplastics counted in stomach contents were fibers, whereas water samples mainly contained fragments. Our results point to microplastic pollution in Lago Menor of Lake Titicaca. However, we could not determine the pollution rate due to considerable methodological limitations. Further research will be needed to robustly detect microplastics in Lake Titicaca and their impact on the fish species in the lake.

Introduction

Plastic is an important component in everyday life (Andrady & Neal, 2009), and its diverse applications, widespread use, particularly for disposable items, has increased drastically (Thompson et al., 2009; Li, Liu & Paul Chen, 2018). Plastic is nowadays considered one of the main pollutants in the environment and has attracted worldwide attention (Thompson et al., 2009; Cole et al., 2011; Eerkes-Medrano, Thompson & Aldridge, 2015). Plastic wastes are observed even in isolated islands in the Pacific Ocean (Lavers & Bond, 2017). Microscopic plastic (i.e., microplastics) is currently a topic of interest to researchers. The plastic characteristics, fragmentation, environmental fate, and potential impacts are the main research topics (Blettler et al., 2018). The term microplastic was coined by Thompson et al. (2004) and further defined by other authors (see Arthur, Baker & Bamford, 2009; Cole et al., 2011; Frias & Nash, 2019; Hartmann et al., 2019). We define microplastic as “plastic particles ranging between 0.1 and 5,000 μm” (Desforges et al., 2014; EFSA Panel on Contaminants in the Food Chain , CONTAM). Microplastic pollution is well-studied in the marine environment, while research on plastic pollution in freshwater environments is still limited (Mani et al., 2016; Blettler et al., 2017; Blettler et al., 2018; Dusaucy et al., 2021). The majority of plastic pollution studies in freshwaters were carried out in Europe and North America, leaving research to do in Asia and Latin America, where progress has been made in recent years (Blettler et al., 2018; Alfonso, Arias & Piccolo, 2020; Galafassi et al., 2021).

The interaction of microplastics and aquatic biota is expected (Silva-Cavalcanti et al., 2017; Alfonso, Arias & Piccolo, 2020). Although microplastic ingestion does not directly impose fatal effects on fish, a negative correlation between fitness and plastic ingestion was observed (Lusher et al., 2014). The pathways of microplastics entering the fish include direct and indirect ingestion. Fishes are the group with most records of plastic ingestion and it is described that fish mainly ingests microplastics via predation activities (Li, Liu & Paul Chen, 2018; Azevedo-Santos et al., 2021). The ingestion of plastics by fish thus may be related to the trophic guild. For instance, omnivorous fish tend to consume more plastic than fish with a specialized diet (Mizraji et al., 2017). Understanding the feeding aspects across fish species facilitates the interpretation of plastic content in the stomach (Dantas, Barletta & Costa, 2015). Freshwater biota is vulnerable to microplastic pollution, particularly fish living in freshwater ecosystems near urban areas (Silva-Cavalcanti et al., 2017; Souza, Corrêa & Smith, 2020). Their vulnerability is caused by the disposal of plastics in freshwater ecosystems coming from direct dumping into rivers or through the disposal of synthetic fiber products from an increasing population and rapid urbanization (Blettler et al., 2018). For example, Lake Titicaca (shared between Bolivia and Peru), one of the highest and most emblematic ancient great lakes in South America (3,809 m a.s.l.), shows a pollution problem that has increased since the 1990s due to poor land-use planning (Molina et al., 2017). The lake is part of the TDPS endorheic system (Titicaca-Desaguadero-Poopo-Coipasa Salt lake), characterized by a north-south gradient, with a single outlet, which drains out to the Lake Uru-Uru and Poopo (Dejoux & Iltis, 1992; Cross et al., 2000; Revollo, 2001). Its physicochemical characteristics are atypical due to its high altitude and tropical location (∼16°S) (Lazzaro & Gamarra, 2014). Lake Titicaca is also the most important water resource for the Andean Altiplano. However, Bolivia’s main source of anthropogenic pollution reaches the lake in Cohana Bay through the Katari River, which drains the densely populated area of El Alto city, with more than 1.2 million inhabitants, industries, and a poor waste water treatment (Molina et al., 2017). Moreover, the pollution on the Bolivian side of Lake Titicaca is exacerbated in the southern shallow basin (Lago Menor), with large areas encompassing depths between 5 to 10 m (Dejoux & Iltis, 1992; Lazzaro & Gamarra, 2014).

Lake Titicaca also hosts a highly endemic biota, dominated by members of the killifish genus Orestias (more than 20 species) and the catfish genus Trichomycterus (2 species) (Ibañez et al., 2014). In addition to the exotic piscivores, rainbow trout (Oncorhynchus mykiss), and silverside (Odontesthes bonariensis) were introduced in the 1940s (Loubens, 1992; Loubens & Osorio, 1992). Although Lake Titicaca provides fish for ∼3 million people in the region the fish stock has been drastically declining, mainly by anthropogenic causes (Ibañez et al., 2014; Monroy et al., 2014; ALT, 2020). A lack of knowledge remains on fish response to pollution. The native and exotic fishes are the base of artisanal fisheries in Lake Titicaca. Most of the native species (i.e., O. luteus, O. agassizii, and T. dispar) predominantly occupy benthic habitats (Lauzanne, 1992; Ibañez et al., 2014; De La Barra et al., 2020) with an omnivorous diet, although with few detailed studies. Conversely, O. bonariensis is a piscivore fish, and is the main predator of native Orestias (Loubens & Osorio, 1992; Monroy et al., 2014). Frequently, the species that are economically important for commercial fisheries are the target of studies on plastic ingestion. However, scarce information exists on microplastic contamination in the fish community of Lake Titicaca. This study, therefore, aims to evaluate the presence of microplastics in the stomach contents of four fishery resources from the Lago Menor of Lake Titicaca caught by local fishermen and to assess whether there is an association with trophic guilds or fishing areas. Additionally, surface water was analyzed to evaluate the presence of microplastics in the water. Finally, our study contributes to the growing body of research on microplastics in freshwater environments.

Materials & Methods

Study area

Lake Titicaca is divided into two basins: the oligotrophic deepest northern basin Lago Mayor (7,131 km2; mean depth = 100 m; max depth = 285 m; water volume = 900 km3), and the smaller southern shallow basin Lago Menor (1,428 km2; mean depth = 9 m; max depth = 40 m; water volume = 12 km3) (Dejoux & Iltis, 1992). The difference between basins in Lake Titicaca make Lago Menor more susceptible to pollution (Molina et al., 2017; Guédron et al., 2017). Similarly, due to its shallow water column, the entire water column of Lago Menor is mixed because of the strong winds (Achá et al., 2018). Therefore, small shallow portions of the lake are now eutrophic (Molina et al., 2017; Achá et al., 2018). Lake Titicaca hosts a strong artisanal fishing activity of considerable importance for the income of rural families. The fisheries in Lago Menor have four main target species (O. agassizii, O. luteus, T. dispar and O. bonariensis), which are caught all year round (Lino & Padilla, 2014; ALT, 2020). Nonetheless, as stated previously, heavily polluted water drains to the Lago Menor through the Katari River from untreated sewage (Molina et al., 2017). This study was carried out in Lago Menor.

Fish sampling

We identified the main landing zones arriving from the same fishing areas, and the rural communities of Huarina, Cachilaya, and Desaguadero were selected (Fig. 1). We followed the fishing activity during the last weeks of May and June 2021 and bought samples composed of 60 to 100 specimens (when possible) of each of the four target species from the fishermen. Often the fish arrived alive at the landing zone, so the entire sample was euthanized in ice, which also reduced digestive action and preserved stomach content. Then, fish were transported in coolers to the laboratory (between one and two hours travel time). Each fish specimen was measured: total length (TL, mm; digital Vernier CD-20CP Mitutoyo) and weight (W, g; PT 120 Sartorius Laboratory precision balance, 0.01 g). After the measurements were taken, the specimens were dissected, and the stomachs were separated. Next, stomach content was washed with distilled water and diluted in case of abundant stomach content. All stomach content was transferred to Petri dishes, where the microplastics were evaluated under a stereomicroscope with a magnification power between 6 X to 50 X (WILD M3, Heerbrugg). Microplastics were identified, counted, and categorized by color and type (fragment and fiber) (McCormick et al., 2016). The visual identification of microplastics may have several limitation, but is still the simplest and most widely used (Li, Liu & Paul Chen, 2018). Our visual evaluation followed personal observations comparing plastic fragments and fibers found in the stomachs with plastic materials found in water samples. In order to prevent any contamination, we used lab coats, gloves, and caps during the evaluation. In addition, all work surfaces and tools were sterilized with ethanol, and all utensils and Petri dishes were carefully assessed before the stomach contents were transferred to ensure the Petri dishes were not contaminated with any plastic. The quantification of microplastics ingested was based on the frequency of occurrence (FO) (Hyslop, 1980), calculated from the equation: FO (%) = Fi/Ft*100; where Fi is the number of stomachs containing microplastics and Ft in the total number of stomachs examined.

Figure 1 Location map of sampling station in Lago Menor of Lake Titicaca.

Map the localization of the Lago Menor in the Lake Titicaca study area. Dark Blue circles show the water sampling replicates along the transect (blue dotted line). Red circles show the three landing zones in Lago Menor, and light blue show the fishing areas near the transect.

Water sampling

The water samples were collected during the survey of the project “Working for a healthier aquaculture in Lake Titicaca” (TEAM Project, VLIR-UOS) in Huatajata, Central Islands, and Desaguadero (July, 2021), and Nacoca and San Jose (August 2021; Fig. 1). Samples were collected along transects from the shoreline to open waters of the Lago Menor. Surface water (between 0.5 to 1 m depth) was collected using a 1 L Van Dorn bottle. Three replicates were taken along of each transect, where 5 L of water was collected in each replicate. The collected water was accumulated in carefully washed gallon jugs prior to use. Then, the 5 L of water was filtered in a 20 µm mesh, concentrated in 50 mL falcon tubes, and stored on ice until it arrived at the laboratory for evaluation under controlled conditions. In the laboratory, the individual 50 mL of water sampled was subsequently suspended with distilled water and filtered once again through a 10 µm sieve to avoid losing any portion of the sample. Next, the sieved material was transferred to a Sedgwick-Rafter counting cell with a glass dropper pipette and visually examined under a microscope (Olympus CX43, 4X, and 10X objectives) coupled to a 3.1-megapixel digital color camera for microscopy (Olympus LC30). The microplastics were identified, counted, and categorized by color and type (fragment and fiber) (McCormick et al., 2016). Finally, 10–15 items of microplastics in fragments and fibers were randomly selected, photographed, and measured using the microscope’s camera software (LCmicro, Olympus Image Analysis Software 2.2 version) without changing any of the image characteristics (i.e., contrast, gain, saturation). The same contamination prevention measures mentioned in the previous section were taken. The Bolivian Ministry of Environment fully approved the development of the project investigations from which this study is derived (MMAYA/VMABCCGDF/DGBAP/MEG 340/2019).

Data analysis

The data presented did not meet the assumptions required for conducting a one-way ANOVA. Therefore, a Kruskal–Wallis test was performed to look for differences in number of microplastics ingested per fish species, trophic guilds and fishing areas. The Kruskal-Wallis test was also used to test differences in the number of microplastics per site. All statistical analysis were performed using Infostat software (Di Rienzo et al., 2017).

Results

A total of 1,283 fish were evaluated, dominated by O. agassizii (32%) (Supplemental Material). We observed microplastics inside the stomach of 44 specimens (3% of the evaluated fishes). Microplastics ingestion was observed in all fishing areas, but none was observed in the stomach of O. agassizii and O. bonariensis at Desaguadero. The ingested microplastics observed in fish were dark-colored fibers (blue and black). The fibers were usually found as filaments (Fig. 2A) and rarely as a bundle (Fig. 2B). The frequency of occurrence of fibers was low in all species (FO <5%; Fig. 3). The carnivorous species (O. bonariensis) showed microplastics at much lower frequency (0.3%). Finally, the mean number of microplastics found per species, trophic guilds and fishing areas were not significantly different (H = 2.64, p = 0.504; H = 1.26, p = 0.297; and H = 4.35, p = 0.111, respectively).

Figure 2 Examples of fibers of microplastics found in gut content.

Photograph of fibers found in stomach content of O. luteus and O. agassizii. (A) Microplastic fiber bun, (B) microplastic fiber with stomach content. The stereo microscope used has a magnification power of 6 X and 50 X (WILD M3, Heerbrugg).

Figure 3 Frequency of occurrence (FO) percentage of microplastics found in gut content of the fish studied in Lago Menor of Lake Titicaca.

Frequency of microplastics fibers found in the stomach content of four target species of Lago Menor artisanal fisheries. Bars indicate the SD.

A total of 3,395 particles of microplastics were counted in water samples ranging between 56 and 602 particles per sample. Microplastic quantity was highest at Desaguadero (103 ± 20 particles per L; Table 1), although there were no differences between sites (H = 7.36, p = 0.118). Of the total number of microplastics identified, 69.4% were blue, 29.4% red, and 1.2% black. The majority of microplastics counted were fragments (54.2%). Desaguadero was the site with the highest quantity of microplastic fragments (75 ±  19 particles per L). Conversely, a higher quantity of microplastic fibers was found in the Central Islands (41 ± 34 particles per L; Fig. 4). The size of particles ranged from 5.63 µm to 2,756.92 µm for fibers and from 2.15 µm to 1,249.34 µm for fragments. The mean size in length of fragments and fibers was larger in Central Islands (fragments: 117.5 ± 183.7 µm; fibers: 645.3 ± 584.7 µm; Fig. 5).

Table 1 Microplastic debris in the water of Lago Menor of Lake Titicaca.

Mean microplastic number per liter debris in the water samples of Lago Menor of Lake Titicaca.

Site	Mean of microplastics counted ± SD	Range of microplastics counted (min-max)	
Huatajata	36.00 ± 00.57	35.60–36.40	
Central Islands	58.87 ± 41.41	11.20–86.00	
Nacoca	23.93 ± 5.82	18.40–30.00	
San Jose	24.60 ± 16.69	12.80–36.40	
Desaguadero	103.13 ± 19.88	81.40–120.40	
Notes.

Number of microplastics found = 3,395 particles (Fibers: 45.8%; Fragments: 54.2%).

Figure 4 Number of microplastics in water samples in Lago Menor of Lake Titicaca.

Box plot for the microplastic number found in the water samples in different sites in Lago Menor of Lake Titicaca. Visual examination was performed using an Olympus CX43 microscope, 4X and 10X objectives.

Figure 5 Size of microplastics found in the water samples in Lago Menor of the Lake Titicaca.

Bar plot of the size in length of microplastics found in the water samples in different sites in Lago Menor of Lake Titicaca. Bars indicate the SD. Visual examination was performed using an Olympus CX43 microscope, 4X, and 10X objectives, and photographed using an attached Olympus LC30 digital camera. No contrast, gain, and image saturation modification to the photographs were made.

Discussion

We observed that 3% of the fish evaluated in our study had ingested microplastics. This low occurrence of plastic particles was much lower than other freshwater fish species reported (Galafassi et al., 2021). Furthermore, the relationship between microplastic ingestion, species, trophic guild, and fishing areas was not evident. However, it is notable that the carnivorous species (O. bonariensis) had the lowest occurrence of microplastics. Conversely, omnivorous species, particularly O. agassizii, had a relatively higher ingestion of microplastics. An important ecological characteristic of this species is its feeding plasticity, which allows it to take advantage of almost all food items in its habitat (Vila, Pardo & Scott, 2007; Ibañez et al., 2014; De La Barra et al., 2020). Specifically, our result may be consistent with the hypothesis that the amount of plastics ingested by omnivorous fish is associated with their wider food web (Mizraji et al., 2017). It has frequently been suggested that fish living in freshwater environments near urbanized regions have a higher risk of microplastic ingestion, and a direct relationship with the level of urbanization has been observed (Silva-Cavalcanti et al., 2017; Souza, Corrêa & Smith, 2020; Galafassi et al., 2021). However, several studies present evidence that higher environmental concentrations of microplastics do not necessarily translate into higher ingestion by aquatic biota (McNeish et al., 2018; Talley, Venuti & Whelan, 2020; Sembiring et al., 2020). Such is the case for our results because the microplastic numbers in the water were higher than in the stomachs. Life history may thus have an even more significant influence on the level of microplastic ingestion (Talley, Venuti & Whelan, 2020). It is important to highlight here that more research is needed to obtain further information on microplastic ingestion by fish in Lake Titicaca. Surely, microplastics can cause health complications to wildlife in aquatic ecosystems. These complications can include intestinal obstruction, gut microbiota, physical injury, and liver stress, among others (Browne et al., 2008; Wright, Thompson & Galloway, 2013; Guerrera et al., 2021; Huang et al., 2022). Furthermore, aspects such as transit time and the ability of the digestive system to get rid of microplastics should be considered in future studies to understand the impact of these pollutants in fish of Lake Titicaca.

The mean numbers of microplastic abundance in freshwater systems vary significantly (Li, Liu & Paul Chen, 2018). Human activities, sampling location and sampling approaches could contribute to differences in microplastic number (Eerkes-Medrano, Thompson & Aldridge, 2015). In our study, the number of microplastics was variable (ranging from 11 to 120 particles per liter), showing higher quantities in the southern part of Lago Menor. Specifically, the highest quantity of microplastics was observed in Desaguadero. It was not possible to differentiate whether this higher number of microplastics is a local phenomenon or is a result of the natural watershed gradient. There is no accurate information on the amount of waste discharged into Lake Titicaca. Moreover, there is no data on the Lago Menor’s water flows (Lazzaro & Gamarra, 2014). The number of microplastics was also higher near the Central Islands, possibly because of the spread of pollution from wastewater inlets into the lake (Duwig et al., 2014; Archundia et al., 2017). The Katari River, the main tributary of Lago Menor, is undoubtedly the main wastewater source in this shallow lake (Molina et al., 2017; Achá et al., 2018). However, physicochemical changes observed in Lago Menor suggest other sources of pollutants, in addition to those transported by the Katari River (Achá et al., 2018). In recent years, demographic growth in the Central Islands in Lago Menor has led to rapid urbanization, with no sewage water or solid waste management, adding to the pollutants discharged into Lake Titicaca. There is a strong link between the rapid growth of urban and industrial areas and the detriment of the water quality of Lake Titicaca, as there is deficient wastewater management (Molina et al., 2017; Archundia et al., 2017; Baltodano et al., 2022). This situation thus highlights the need for a monitoring system for Lake Titicaca, one of the 20 largest lakes in the world.

It is worth mentioning that the methods used to evaluate water and the stomach content of the collected fish were based on visual observations and could therefore be biased. Furthermore, none of the samples studied were subjected to a purification process nor density separation. Therefore, our results should be taken cautiously as no analytical methods was performed. Although the current techniques for quantitative and qualitative research of microplastics include spectroscopy and liquid chromatography (Li, Liu & Paul Chen, 2018), the availability of this specialized equipment is not always easily accessible to researchers in developing countries. Unfortunately, our study has methodological limitations and does not meet all the requirements and quality assurance/quality control proposed by Cowger et al. (2020). Specifically, visual observation of plastics can be misleading, and, more importantly, the calculated number of microplastic presence could be an underestimate or overestimate of the real number. In addition, the results of visual classification are strongly affected by several factors (i.e., carelessness of the assessor or the quality of the microscopy) (Hidalgo-Ruz et al., 2012). To date, there is no information on microplastic pollution in Bolivia or Lake Titicaca (Castañeta et al., 2020; Galafassi et al., 2021). Therefore, despite its methodological limitations, our study provides evidence of the pollution of microplastics in Lake Titicaca’s habitat and fishery resources.

Conclusions

In this study, we report the presence of microplastics in the stomachs of the four target species of artisanal fisheries in the Lago Menor ok Lake Titicaca. Our results suggest that the frequency of microplastic ingestion by fish is low, and was not related to the trophic guild or fishing areas. However, the presence of microplastics in the water was particularly high in Desaguadero, near the outlet of Lago Menor. The difference between microplastics found in stomach contents and the water column suggests that microplastic pollution and urbanization were not related in Lago Menor. However, we cannot ensure the pollution rate in Lago Menor due to considerable methodological limitations. Nevertheless, despite these limitations, this study represents relevant information assessing the microplastic content in the emblematic Lake Titicaca. Further research would be needed for more robust detection of microplastic quantity for Lake Titicaca fishes across large spatial scales.

Supplemental Information

Supplemental Information 1 Microplastics countings in water and fish samples

The count of microplastics found in the gut contents analysis of the fishery resources evaluated; the counts of microplastics found in the water samples; and the subsamples of microplastics found for measurement.

Click here for additional data file.

Supplemental Information 2 Number of specimens analyzed, habitat, feeding habits and diet, body length, and weight ranges of the sampled fish from Lago Menor in Lago Menor of Lake Titicaca

Click here for additional data file.

The authors would like to thank to Gisela Sarzuri and Fanny Nina for their support in stomach content evaluation, and to the coordinator of the GIRH-TDPS project of the UNDP, IPD-PACU and IMARPE for allowing the use of partial information concerning fisheries resource of Lake Titicaca for this publication. In addition, we would like to thank the fieldwork team of “Working for a healthier aquaculture at Lake Titicaca” project (Viviana Cruz, Debora Alvestegui, Adilen Fernandez and Jorge Molina) for their support for the water sampling collection.

Additional Information and Declarations

Competing Interests

Author Contributions

Animal Ethics

Field Study Permissions

Data Availability

The authors declare there are no competing interests.

Erick Loayza conceived and designed the experiments, performed the experiments, analyzed the data, prepared figures and/or tables, authored or reviewed drafts of the article, and approved the final draft.

Amaya C. Trigoso Barrientos analyzed the data, prepared figures and/or tables, authored or reviewed drafts of the article, and approved the final draft.

Geert P.J. Janssens conceived and designed the experiments, prepared figures and/or tables, authored or reviewed drafts of the article, and approved the final draft.

The following information was supplied relating to ethical approvals ({i.e.}, approving body and any reference numbers):

The Bolivian Ministry of Environment fully approved the development of the project investigations from which this study is derived (MMAYA/VMABCCGDF/DGBAP/MEG/ 340/2019).

The following information was supplied relating to field study approvals ({i.e.}, approving body and any reference numbers):

The Bolivian Ministry of Environment fully approved the development of the project investigations from which this study is derived (MMAYA/VMABCCGDF/DGBAP/MEG 340/2019).

The following information was supplied regarding data availability:

The raw data are available in the Supplementary Files.

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
