# Peer review of "Evidence of microplastics in water and commercial fish from a high-altitude mountain lake (Lake Titicaca)"

_PeerJ, doi:10.7717/peerj.14112_

## Round 0.1 · original submission · Major Revisions

Three experts have assessed your manuscript and identified a number of issues that make the manuscript unacceptable in its present form.

The two most important aspects are the identification of microplastic particles and the lack of proper quality control and assurance measures (QC/QA). You applied a visual identification of the particles. As pointed out by the reviewers, this shortcoming may result in a high rate of misidentified particles. As a minimum requirement, you should address the limitations of your applied visual identification approach, or – even better – apply a spectroscopical (FTIR or Raman spectroscopy) or a mass-based proof of the particles (pyrolysis-GC/MS) in case you have retained samples from your study. Regarding QC/QA measures a minimum requirement are procedural blanks to control for microplastic contamination by sample handling.

On the other hand, the reviewers underlined the unique position and great importance of your study, so I hope that their criticisms will allow you to carry out a substantial revision of the manuscript, which is a precondition for acceptance of the manuscript.

·

Basic reporting

The manuscript is written in a clear and in general understandable English. The structure is reasonable and follows a clear logic. No hypotheses are given.

The Introductory section refers to the most relevant literature and indicated specific characteristics of the study site, namely the high altitude and correspondingly the high UV radiation. The aims of the study are not explicitly stated, nor are hypotheses proposed. Also, it is unclear why data on the color of microplastic were raised.

The literature is well referenced, however, not all papers are relevant in this context (e.g. van Cauwenberghe et al. 2013, Pervez et al. 2020), and some references are outdated and superficious (Hoss et al. 1990; Carpenter et al 1972). Further, there are a couple of excellent reviews and books on this issue, e.g. Triebskorn et al 2019 (in: Trends in Analytical Chemistry), Wagner & Lambert 2018, and many more, so, the authors should consider those as background literature.
The literature on microplastics in fish gut shows a very wide spectrum of microplastic contamination, ( 0-0.25% up to 30%). For comparing the own results with what was published, the authors should have included more relevant papers, especially when they discuss the relation between water surface contamination and ingested microplastic in fish, as well as the importance of feeding behaviour

Experimental design

The research is original and new, it is in the Aims and scope of the journal. However, the research questions are not well defined. Most importantly: The methods do not qualify this manuscript for publication.
The authors assessed the identified particles and fibers visually. Meanwhile, it is standard to provide a thorough, state of the art analysis of the plastic particles found in the environmental samples. To verify the chemical composition of the found particles, fragments, fibers, chemical-analytical proof of polymers has to be performed (e.g. FTIR, Raman spectroscopy, etc.). The use of appropriate methodology for microplastic confirmation, i.e. identification of polymer type is emphasized in many leading papers (e.g. Hermsen et al., 2018, Koelmans et al. 2019) since even experienced microplastic researcher strongly overestimate the proportion of microplastics (cf: Bergmann et al 2017: visual misidentified particles 83%).

Also, no quality assessment and quality control (i.e. positive and negative controls) were applied. Studies investigating environmental microplastic pollution are susceptible to sample contamination, and thus strict counter measures have to be adopted. Also for these and other aspects, high quality papers on the standards with respect to sampling, laboratory preparation, sample treatment etc. are to be consulted and followed prior of designing the study.

These flaws are so severe – and so fundamental (!) - that I cannot approve a publication of this manuscript.

Validity of the findings

Clearly, the manuscript fails to meet the standards on the state-of-the art microplastics investigations of environmental compartments. This is a pity, given the huge work which was performed. In case the particles and fibers are still available, they should be analysed (see above).

·

Basic reporting

The study provides a report on microplastics in water samples and the intestines of fish samples acquired from local fishermen from Lago Major of Lake Titicaca. I see major gaps in reporting and quality control and would therefore recommend a major rework of the paper before it can be considered for publication.
General comments
It is noteworthy that the cited literature across the whole paper (54 references in total) is mostly older than 5 years (83 % of citations <= 2017). I see this as a disconnect from the rapidly developing field of microplastics research and the striving of the community to improve on reported information and quality control standards for monitoring studies. Here, Cowger et al. (2020) are a good reference on what information should ideally be provided.
Abstract:
• L22: fishery species -> fish species
• L24: missing unit (particles L-1)
• L28f: “even into smaller pieces” -> “into even smaller pieces”
• L29: nanoplastics or smaller? What do you consider smaller than microplastics?
• L29: “understand better” -> ”better understand”
Introduction:
As mentioned earlier, the introduction shows a disconnect from methodological and quality control standard development efforts over the last couple of years. No mention of sampling methods and missing references for relatively strong statements:
• Microplastics sizes between 0.1 µm and 5 mm (No source, there is no consent on that; L23; (Hartmann et al. 2019))?
• Impact on human health (L35)?
• Potential for MP to accumulate in tissues (L56)?
Other comments:
• L34: of global concern
• L40: poor wording that something inherently anthropogenic like plastic is released from the terrestrial environment.
• Why mention marine at all if your study is limited to a freshwater ecosystem?
• L42: poor wording: “their study for freshwater systems is still limited”
• L49: “Ingestion of plastics by fish is not new”. So? What do you want to tell us?
• L50f: Is that true for fish as well? Are fish not more selective? There are studies on fish certain species having a preference for blue MP because they resemble their usual prey animals. Also, you’re not coming back to such a point in the discussion.
• L57: So far, literature does not indicate that particles accumulate much, so concepts like bioaccumulation and biomagnification should be applied with caution. When speaking of associated contaminants you make no distinction between material-borne substances such as additives and chemicals of other origin that accumulate in plastics.
• L59: Plastics in fish. Does that mean occurrence of plastics in fish species in the wild? I believe this reference is not up-to-date.
• L60f: “Its biota…” This phrasing doesn’t make sense.
• L75: “Lago Menor’s human activities”. The lake doesn’t have human activities.
• F86ff: Poor phrasing.
• F92: Repetition of sentence.

Experimental design

Materials & Methods
• Study area information would be more fitting in the introduction but is also not really picked up on in the discussion
• Paragraph following L113: That is also not discussed or picked up on later on, so what is the relevance to the paper?
• Paragraph following L122: If human exposure through food fish is your concern, isn’t aquaculture then not relevant both as a source and as an exposure pathway? Also, most information in this paragraph is also more introductory than about the actual study.
• Fish sampling: Landing zone does not equal sampling area. Do you have information on how far from those landing zones fish are collected?
• Throughout the manuscript: the plural of specimen is specimen, not specimens.
• L137: The fish were euthanized in ice? Were they not dead yet?
• Was there no gut content that obscured the view?
• How did you measure fiber sizes/lengths?
• Did you use any size cutoffs? How did you select the MP you measured? How did you prevent contamination? Do you have any thoughts about the origin of mainly black and blue fibers? That could easily be a contamination from clothing. Under what conditions did you investigate the samples apart from cleaning the petri dishes?
• The Figure 4a shows a bundle of fibers. Was that a regular occurrence? Did you just untangle such bundles and measure the individual fibers?
• Water sampling: Why did you choose this depth? How does this fit into other data on depth profiles? Three replicates of water samples à 1 L, but five L per replicate? That is confusing.
• L152: Five L was -> were.
• L152 says filtration through 20 µm, but L156 says filtration through 10 µm sieve. Did you really store 5 L of water in 50 mL falcon tubes? Why?
• L160: How did you select those 10-15 MP and what did you measure and how?
• L164: Poor phrasing throughout the paragraph. (as well as -> and)

Validity of the findings

Results
• Information on the fish measurements are not relevant to your results unless you plan on normalizing MP ingestion to body size or wet weight.
• More interesting: is it normal for fish to have completely empty stomachs? Or could that be something that happens during the catch? What kind of light does this shine on your results?
• Again, what measures did you take to prevent contamination of samples from air and clothing of the involved researchers.
• Was there any gut content? How did you deal with it? What part of the gut was observed, how was it removed? Was it removed?
• How do you know your measured subsample of particles is representative?
• How do you determine fiber length? What dimensions do you measure about a particle? How do you define a Fiber?
• Raw data (MPinGut): Desaguadero O. luteus 45 1 45; this adds up to 46, not 45
• Raw data (MP counting): 1SJ3S 1 SAN JOSE 3 20/8/2021 SURFACE 5 114 68 407
o This adds up to 182, not 407. Why the difference?
Discussion
• L194: Is plastic production more relevant for plastic emission or plastic use?
• L197f: If this is the sole goal of the study then you could shorten it significantly.
• I would appreciate a more critical discussion of sampling and analytical methods, especially related to visual-only identification and its boundaries.
• L216 since 2012 and 2018 leaps have been made in the discussion on this topic.
• L220ff: How does this relate to the kind of particles of you found? Are certain sources more likely to emit such fibers? Any thoughts on materials? What about fishing gear or fisheries-related contamination?
• L225: This paragraph is not well structured and UVR in my opinion would not be the most important point of discussion here. As you do not discuss size cutoffs for your methods why bother with discussing fragmentation? Material degradation is not part of your paper and you do not provide information on the weathering state of the particles.
• How would the sampling method influence your findings? How are the fish caught? Would it make sense to you to find fibers in such high abundance in relation to potential entry sources? This would be the point to discuss geographical peculiarities about the lake and sampling sites and critically reflect on information gaps.
• The question remains why only so few fish actually had MP in their guts, while the water is so highly polluted.
Conclusion
• L242f: Is lack of data by itself a sufficient motivator?

Additional comments

References

Cowger, Win, Andy M. Booth, Bonnie M. Hamilton, Clara Thaysen, Sebastian Primpke, Keenan Munno, Amy L. Lusher, et al. 2020. “Reporting Guidelines to Increase the Reproducibility and Comparability of Research on Microplastics.” Applied Spectroscopy 74 (9): 1066–77. https://doi.org/10.1364/AS.74.001066.

Hartmann, Nanna, Thorsten Hüffer, Richard C. Thompson, Martin Hassellöv, Anja Verschoor, Anders Egede Daugaard, Sinja Rist, et al. 2019. “Are We Speaking the Same Language? Recommendations for a Definition and Categorization Framework for Plastic Debris.” Environmental Science & Technology. https://doi.org/10.1021/acs.est.8b05297.

·

Basic reporting

• The English language should be improved. As example see line 43-44, 51, 56-57, 64-66, 92, 200-201, 204-206, 207-209, 231-232 and more. Please rephrase the sentence in line 246-248, I do not understand the meaning.
• Several passages implying causal relations where there seem to be none (see line 34-36, 36-38, 59-60, 87-89, 94-95, 240-241 and more). Please pay attention on this and rephrase the respective sentences.
• The manuscript provides background information on the current knowledge on MP occurrence in freshwater ecosystems and mentions some sources (line 40-45, 63-70). This ccould be improved by providing more detailed information on general microplastic (MP) sources and known sources of pollution of the Lake Titicaca and better structuring this information. Further,
o please provide more details on recent publications on the impact on MP exposure to fish (line 47-59). Please provide a source for the statement on line 58-60.
o Please give further details on why the non-compliance of fishery management measures mentioned (line 89-94), is relevant in this context and requires the assessment of MP in the lake (as stated in the text line 94-95).
o Please consider removing line 73-74 (photoinhibition phytoplankton), as it does not seem to be relevant for this study.
• Further, the “Material and Methods” part is very long (800 word + Fig. 1), but relevant information is missing (see comments in “Experimental design”). I suggest shortening the “Study area” part and rethink the relevance of the presented information for this study. For example, the details about the lake’s ecosystem (line 118-126 and 130-136) are not being picked up on in the results/discussion.
• Results:
o The reported fish size and weight is not being discussed, no link of fish size/weight to microplastic occurrence or similar is being made. I suggest having this information (line 179-185 and Tab. 1) in the supplementary information.
o The raw data on MP occurrence is provided (small error in tab MPinGut, Column FO, row 2). No raw data on the fish parameters are provided.
• Tables and Figures:
o Table 1: What is N and why are there 2 numbers per species?
o Table 2: Please mention which “mean and SD“ you present in the table.
o Please mention the samples size/replicates in all graphs.
• Within line 240-246 there is some repetition, please adjust.

Experimental design

• In the manuscript it is stated which research gap is being addressed by this study (scarce/missing information on MP occurrence in South America/Bolivia) and how this study will fill the gap (investigating the occurrence of MP in Lake Titicaca and 4 fish species). In this context, please specify line 98-99. What is “general information about the status of microplastic pollution” and is this being addressed by the method? Further, please clarify why this study can be “a baseline for the assessment of sources”, have the sample sites been chosen accordingly?
• The relevance of this study is mentioned as well (general MP problematic and importance of the lakes freshwater and food recourses). The Last paragraph of the introduction can be improved by rephrasing and reordering some of the information as mentioned in “Basic reporting”.
Fish sampling:
• The sample size of the analyzed fish is large, but there are several shortcomings in the study design and methodology (see Hermsen et al. 2018 and Wang and Wang 2018 for reference).
• Very limited quality assurance and control (QA/QC). The only reported quality assurance was rinsing of the petri dishes. No procedural blanks to control for MP contamination by sample handling.
• No digestion, separation or filtration of the gut content prior to analysis of MP. The chance of missing particles is very high, but not described (no recovery study).
• MP detection by visual examination of gut content under a stereomicroscope. No further (spectroscopical or thermo-analytical) identification of the particles, to confirm that the particles are synthetic polymers. No recording of the particles size.
Water sampling
• No quality assurance and control (QA/QC). No procedural blanks analyzed to control for sample contamination (for example via the plastic tubes used for storage of the water samples). No (size) limit of detection is reported.
• Detection of MPs by visual inspection with an optical microscope. No further identification (see above) makes the results prone to bias, especially for the smaller size range (under reporting).
• Line 167, 168 by “measuring”, do you refer to size? How did you base your selection of the 10-15 particles to be measured (line 167)?
I suggest reanalyzing a randomized subset of the identified particle (from water and biota) with a spectroscopical or thermo-analytical method to confirm their identification as MP. Please provide these additional data.
A large improvement would also be the reporting of a limit of detection (quantity and size). Please consider to assess these limits.
In “Material and Methods” important details about the sampling and analysis are missing. Please provide these details:
o Which fish species were investigated?
o Fish sampling frequency and sample size (total and per species) (line 140-143).
o Could you please expand on the sampling method (line 142-143).
o Was the fish stomach or gut analyzed (line 22 and raw data vc. 146 etc.).
o Coordinates of water sampling sites. Reasoning for choosing the sites.
o Can you please expand on the water sampling method? How often did you take samples? How do you define a replicate? Is it a sample taken at different dates? What are the stations you mention in the text? In in the raw data file, you mention site and station.
o Concerning the filtering, what kind of 20μm mesh did you use and what kind of sieve?

Validity of the findings

• The main findings, the presence of MP in water and fish of Lake Titicaca are supported by the collected data. The reported frequency of detected MP in the fish can be underestimated due to limitations in the methodology, conclusions are difficult to draw. Please mention this limitation.
• The reported differences in MP fibers and fragments between water and biota, can be the result of a methodological bias. The fish samples have not been cleaned; thus, it might be that only the easier to detect fibers were found in the fish sample. Whereas the water samples most probably had less organic material, making the detection of fragments easier. Please discuss these limitations.
• The data on size distribution of MP in the water samples must be taken with caution, without knowing how the analyzed particles have been selected. Please mention possible underreporting of smaller sizes, due to increased difficulty to visually detect them.
• The detection of only blue, red and black particles might be due to the increased difficulty to visually distinguish and detect plastic particles of other colors.
• Line 217 -218, please expand on how this can help reduce MP ingestion or remove this part.
Conclusion
• Line 260-262 the generation of nanoplastics and presence in zooplankton has not been analyzed or discussed in this study, please remove it from the conclusion.
• Please specify your last sentence (line 263-265), it is very generic.

Additional comments

A study out about MP occurrence in Lake Titicaca can benefit this research field and the main finding is supported by the data, but the mentioned methodological weaknesses and the lack of proper quality control and assurance undermines the quality of the data collected. Especially the lack of procedural blanks reduces the quality of this study. I propose this manuscript for publication only under the condition of collecting additional data that increase the robustness of the results.

---

## Round 0.2 · Minor Revisions

Thank you very much for your considerable efforts in revising the manuscript. Although the quality of the manuscript has improved, there are still a number of issues which have to be resolved in a further revision. These are clearly named by both reviewers. I acknowledge that the methodological limitations mentioned - namely the purely optically based identification of the plastic particles - cannot be satisfactorily resolved in retrospect. So this is not a reason for me to reject the manuscript, especially since you refer to these limitations in the revised manuscript expressly and discuss them appropriately. This aspect no longer has to be taken into account in further revisions. I expressly point out the possibility of using the editing service offered by PeerJ to also improve the language of the manuscript.

·

Basic reporting

-As commented earlier the language could use some work, but does not hinder in the understanding of the research.
-Earlier comments hinted at including more recent references, since the field of microplastics is rapidly advancing and the body of literature is growing at a fast pace, so basing a large part of the paper on a reference from 2018 (Li, Liu & Paul Chen, 2018) is not ideal.
-Using abbreviations in the abstract is not good practice and should be avoided, especially if they are not introduced in the abstract (FO is only introduced in the Materials & Methods, but used in the abstract).
-The introduction could be shortened significantly. A lot of the geographical information is not useful in the context of the paper since it is not further elaborated on in the discussion
-The biggest improvement of the work, in my opinion, can be noted in the Results and Discussion sections

Experimental design

Research questions have been more defined, limitations of the design have been adressed.
Method description has been expanded.

It might make sense to calculate the body burden/particles found per specimen for the different landing zones instead of a binary descriptor (yes/no for the number of investigated guts), but this will likely not change the results.

Validity of the findings

I still have doubts about the approach of obtaining specimen from fishermen that have aparently been in the net for an unclear time and that to some degree have still been alive at the moment of purchase, so it in unclear if regurgitation of gut content may have occured and may explain the low frequency of observation of microplastics and to some extend gut content whatsoever. But I suppose those are questions that cannot be answered in hindsight.

Additional comments

I would like to appreciate the high number of specimen that has been investigated, a lot of work has been put into this study and I congratulate on including the raw data!

·

Basic reporting

I appreciate that the authors have responded and addressed most of the feedback form the reviewers.

Relevant background to place this study in context is provided, highlighting the gap that this study fills by studying the occurrence of MP in Lake Titicaca.

The English language improved clearly but should still be worked on. As example see:

- line 22: "four-target fisheries" I suggest: "in four fish species targeted by local fisheries..."
- line 27-29: Please expand on: "differences were not found". Differences between what?
- further please check: line 29-30, 67-86, 144, 169,
- please rephrase: line 33-34, 44-45,69-70, 78-80, 256-257
- The part about the definition of MP (line 44-51) can be shortened/rephrased
- line 57, is "biota" fitting better than "biodiversity" here?
- line 71: Why was "increasing" written here?
- line 76-77: "...close to urbanized areas." are impacted?
- line 214: size in length? Please add.
- line 247: I suggest adding: "reported" microplastic numbers
- line 295: please rephrase: "no relationship was observed between microplastics, ..."

Tables and Figures:
Tab. 1: I assume this shows mean MP numbers per liter? Please add to description.
Fig. 2: please add a scale
Fig. 4 and 5: Something got mixed up here. Description does not fit to labeling, size and particles per liter.

Supplementary information has been provided.
A small calculation error in tab "MPinGut", first row (5.7% should be 5%)?

Experimental design

A large number of samples has been analyzed, unfortunate the visual identification of MP does not meet the standards of the field and the quality control and assurance conducted is not sufficient.

The research question (presence of MP, impact of feeding habit and fishing site) is stated but can be expressed clearer (referring to "look for relationships"). Also, line 120: why landing zones and not fishing areas, this is where the water samples have been collected (mostly). Further, having the labeling according to fishing areas makes it easier to follow (comparison to water sampling sites).

Validity of the findings

Due to methodological limitations (visual identification, insufficient QC/QA) the data is not robust.

To a large extend the conclusions are linked to the research questions.

A link between MP occurrence in the water samples and the fish samples is discussed, but this is not addressed in the results section. See line 233-234: I believe this reasoning is not supported by the fact that MP numbers were higher in water than in fish. Was there a correlation (or missing of such) between sites with higher MP concentration and MP content in fish caught in these areas? Was this assessed?
- line 182: please mention which categories you refer to?
- line 202-205: please rephrase/expand on what was analyzed exactly. Are you refering to no significant differences?
- line 208: no significant differences?
- line 232: higher ingestion by?
- line 251: Which "local phenomenon" are you referring to?
- line 265-267: please rephrase. Before and after this sentence the focus lies on deficient/non existing wastewater management. So it is not clear what is referred to with "these shortcomings" when you write about wastewater treatment plants as MP source.
- line 267-268: repetition. Please relocate/rephrase.
- line 271: please expand on what you refer to? Different sizes comparing water and fish samples, different sampling sites?

Additional comments

I appreciate that the authors have responded and addressed most of the feedback form the reviewers, including a discussion of the limitations. The quality of the manuscript has improved substantially.
Unfortunately, no additional identification (spectroscopical, thermo-analytical methods) of the detected particles could be performed or quality control and assurance (procedural blanks) could be provided (see first review). Unfortunately, without this data, I cannot recommend a publication of this manuscript. The data is not robust enough to address relevant questions raised in the manuscript.

---

## Round 0.3 · accepted · Accept

Thank you for the revision of the manuscript, in which you adequately considered all the reviewer's suggestions. As editor I have decided to accept your submission despite the fact that there are methodological deficits in the identification of the plastic particles, as already mentioned by the reviewers in the two revision rounds. On the other hand, you discuss these limitations sufficiently. Also, since there is little data on microplastics in freshwater ecosystems the Neotropics, the work should be published. There are still some minor linguistic issues that should be improved in the manuscript, but these can be corrected as part of the production tasks.